# Pharmacological Activities and Characterization of Phenolic and Flavonoid Compounds in *Solenostemma argel* Extract

**DOI:** 10.3390/molecules27238118

**Published:** 2022-11-22

**Authors:** Rafaat M. Elsanhoty, Mohamed S. M. Soliman, Yehia A. Khidr, Gamal O. O. Hassan, Ahmed R. A. Hassan, Mohammed Aladhadh, Asmaa Abdella

**Affiliations:** 1Department of Industrial Biotechnology, Genetic Engineering and Biotechnology Research Institute, University of Sadat City (USC), Sadat City 32897, Egypt; 2Department of Plant Biotechnology, Genetic Engineering and Biotechnology Research Institute, University of Sadat City (USC), Sadat City 32897, Egypt; 3Medicinal and Aromatic Plants Department, Natural Products Unit, Desert Research Center, 1 Mathaf El-Matarya St., El-Matareya, Cairo 11753, Egypt; 4Department of Food Science and Human Nutrition, College of Agriculture and Veterinary Medicine, Qassim University, Buraydah 51452, Saudi Arabia

**Keywords:** *Solenostemma argel*, antioxidant, anti-inflammatory, anticancer, antimicrobial

## Abstract

*Solenostemma argel* is a desert medicinal plant indigenous to African countries. This research aims to study the pharmacological properties of *Solenostemma argel* plant. Aerial parts (leaves and flowers) of *Solenostemma argel* (Delile) Hayane were tested for antibacterial activity, antioxidant activity, anticancer, and anti-inflammatory activity. Phenolic and flavonoid contents of the plant were characterized. There was an increase in the antioxidant activity of *Solenostemma argel* extract from 12.16% to 94.37% by increasing concentration from10 µg/mL to 1280 µg/mL. The most sensitive organism was *S. epidermidis* with chloroform extract. The MTT assay revealed that methanolic extracts of *Solenostemma argel* showed potent cytotoxic effects on the A549, Caco-2, and MDAMB-231 cell lines, respectively. The anti-inflammatory activity increased by increasing the concentration of methanolic extract of *Solenostemma argel*, using indomethacin as a standard. Gallic acid was the most abundant phenolic acid, followed by synergic acid and p-coumaric acid, respectively. Catechin, quercetin, luteolin, kaempferol and rutin flavonoids were also found in the methanolic extract. GC-mass analysis showed that aerial parts of *Solenostemma argel* were rich in 2-(5-methyl-5 vinyl tetrahydro-2-furanyl)-2-propanol (11.63%), hexanoic acid methyl ester (10.93%), 3-dioxolane,4-methyl-2-pentadecyl (9.69%), phenol, 2-(1,1-dimethylethyl) (8.50%). It can be concluded that *Solenostemma argel* methanolic extract contain natural bioactive constituents with potential medicinal importance such as antioxidants, antimicrobial, anti-inflammatory, and anticancer activities.

## 1. Introduction

The use of medicinal plants for therapeutic purposes is an ancient form of medication, which is universal among most non-Western countries and predates modern medicine [1]. It is thought that a significant percentage (~75%) of the human population uses herbs and other forms of traditional medicines to treat diseases [2]. In addition, a substantial number of current pharmaceuticals such as aspirin, codeine, and quinine used for disease therapeutics in modern medicine are derived from historical herbal remedies and medicinal plants [2,3]. Therefore, medicinal plants are important sources of highly effective pharmaceuticals for disease treatment and sustainable human health [3].

Most common human diseases are caused by microbial infections and cellular damages caused by oxidative stress related to imbalances between the formation and neutralization of pro-oxidants [4,5]. Oxidative stress, caused by free radicals alongside lipid peroxidation, is a contributory factor to multiple diseases including cardiovascular diseases, atherosclerosis, cancer, and inflammatory diseases [6,7]. Anand et al. [8]; Marrelli, [9] showed that some medicinal plants can be used to treat diseases linked to pathogenic micro-organisms and cellular damages such as inflammatory and immune disorders. Medicinal plants contain primary or secondary metabolites such as lignans, flavonoids, terpenoids, tannins, phenolic acids, alkaloids, quinones, and coumarins, which exhibit significant antioxidant, antimicrobial, and anti-cancer activities [8,9,10].

Cancer is one of the main causes of morbidity and mortality worldwide [11]. Cancer is caused by unregulated cell proliferation, which leads to the disruption in the function and structure of surrounding tissues [12,13]. There are more than 100 forms of cancer with prostate cancer (in men) and breast cancer (in women) being the major forms of cancer in humans [14,15]. Despite the availability of treatments such as radiation, chemotherapy, and immune and hormone therapy, cancer remains one of the leading causes of death worldwide aside from cardiovascular and infectious diseases [13]. While some of these cancer treatment approaches are moderately effective, their long-term use and success rates are hampered by associated chronic toxic effects [13]. Given that free radical activities and inflammation are associated with cancer development, drug candidates that can scavenge free radicals and inhibit inflammation with limited toxic effects are better suited for use as anticancer agents.

Therefore, medicinal plants continue to be investigated as a resource to treat different cancers such as colon cancer [16], breast cancer [17], and prostate cancer [18]. Nevertheless, no silver bullet is yet to be found from medicinal plants for the complete treatment and cure of cancer despite the evaluation of many candidates. Hence, there is a need to continue the evaluation of more plants for their anti-cancer and anti-inflammatory properties that would inhibit or prevent the development of cancer. One of the promising candidates is a plant called argel.

Argel (*Solenostemma argel*) is a wild herbal plant widely found in different countries such as Sudan, Saudi Arabia, and Egypt. Different parts of this plants (leaves, stems, and bark) are used for pain relief and treating infections affecting the respiratory, urinary, and gastrointestinal systems [19]. These parts have also been used to treat liver, kidney, diabetes, and cardiovascular diseases in these countries [19]. Pregnane and phenolic glycosides extracts from the plant’s aerial and root parts have been shown to possess antioxidant properties in human tumor cell lines [20]. Acetone extracts of this plant have also been shown to have antioxidant and anti-inflammatory properties in laboratory rats [21]. In spite of Solenostemma argel widespread use and numerous properties, very little research has been carried out to evaluate its therapeutic properties. The purpose of this study was to assess the anticancer, antibacterial, anti-inflammatory, and antioxidant activity of different extracts from *Solenostemma argel*, as well as to conduct GC-mass and HPLC analyses of volatile components and phenolic acids.

## 2. Materials and Methods

### 2.1. Chemicals

All chemicals and HPLC standards used were of high analytical grade, purchased from Sigma Chemical Co. (St. Louis, MO, USA).

### 2.2. Plant Materials and Sample Preparation

The plant materials, *Solenostemma argel* (family Apocynaceae), used in this study were collected from Halaib and Shalatin (Gabal Elba) region in Egypt. The collected plant materials were washed using distilled water. The aerial parts were cut and air-dried in the shade. These air-dried plant materials were ground into a fine powder using Moulinex Dpa241 Choppers, Blender 1000, Ecully, France, and stored at −20 °C in screwed-in brown bottles until needed for further investigations.

### 2.3. Extraction Process

One hundred grams of the powdered *Solenostemma argel* (aerial parts) was extracted by shaking at 150 rpm for 24 h at 25 °C with 1 L of solvent (water, methanol, acetone, ethanol, chloroform, ether, ethyl acetate, and methylene chloride). The extracts were subsequently filtered using a Buchner funnel containing a Whatman No. 1 filter paper. The obtained residue was re-extracted with 500 mL of solvent, and filtered, after which the filtrates (extracts) were pooled together and concentrated using a rotary evaporator at reduced pressure (Heidolph VV 2000,Hei-VAPCore, Schwabach, Germany. The concentrated extract was dried in a desiccator under a vacuum until a consistent weight was obtained. The weight of the triplicate extracts (samples) was recorded, and each extract was re-suspended in the smallest amount of solvent possible to achieve a concentration of 10 mg/mL. The extracts were stored at −4 °C until use.

### 2.4. Antioxidant Capacity Assay

Antioxidant activity was determined using 2,2-diphenyl-1-picrylhydrazyl (DPPH) assay as described by McCue et al. [22]. The assay reaction mixture contained 40 μL of *Solenostemma argel* extract from different solvent extractions at different concentrations, ranging from 10 to 1280-μg/mL, which had been prepared by diluting the extract with the extraction solvent and 3 mL of methanolic solution of 0.1 mM DPPH radical. The mixture was vigorously agitated and incubated at 37 °C for 30 min. The absorbance values were measured at 515 nm using a UV–visible spectrophotometer (Milton Roy, Spectronic 1201, SpectraLab Scientific Inc., Markham, ON, Canada). Ascorbic acid was used as a positive control. The absorbance value of the reaction mixture was calculated using the following equation:DPPH scavenging activity (%) = 100 × (A_0_ − A_1_)/(A_0_)(1)

The absorbances of the control and sample, respectively, are A_0_ and A_1_. The results are presented as the average of three replicate analyses, with the major values as well as the standard deviation (SD) provided.

The 50% inhibitory concentration (IC50), the concentration required to 50% DPPH radical scavenging activity was estimated from graphic plots of the dose–response curve using Graphpad Prism software (San Diego, CA, USA).

### 2.5. Antimicrobial Activity

#### 2.5.1. Microorganism

Eight clinical microbial isolates were used in the experiments (Gram-negative bacteria: *Escherichia coli*, *Pseudomonas aeruginosa*, *Enterobacter cloacae*, and *Acinetobacter baumannii*; Gram-positive bacteria: *Staphylococcus aureus*, *Streptococcus epidermidis*, and *Enterococcus faecalis* and yeast: *Candida tropicalis*. The isolates were collected and identified from clinical specimens (sputum, end tracheal tube (ETT), nasal swab, and laryngeal swab (from respiratory tract infections patients) from Giessen University Clinic, Giessen, Germany [23].

#### 2.5.2. Paper Disc Diffusion Assay

For disc diffusion assay, 100 µL of cell suspension was uniformly spread onto Mueller Hinton agar (MHA). The 6 mm diameter filter paper discs (Whatman No. 41) were positioned to contact the surface of infected agar and were impregnated with 25 µL extract at a concentration of 10 mg/mL. The plates were incubated for 24 h at 37 °C. The diameter of the zone of inhibition (ZOI) was then measured precisely, and the means of the triplicates were determined [24].

#### 2.5.3. Determination of Minimal Inhibitory Concentration (MIC) and Minimal Bactericidal Concentration (MBC)

MIC and MBC of different *Solenostemma argel* extracts (water, methanol, acetone, ethanol, chloroform, ether, ethyl acetate, and methylene chloride) against all isolates were determined. Briefly, the two-fold dilution method by Koo et al. [25] was used with a concentration range between 3.12 and 100 mg/mL (*w/v*) in Mueller Hinton broth. The tested bacterial inoculum load was 5 × 10^5^ cfu/mL. Each microbial isolate was inoculated into all the dilutions and the inoculated tubes were incubated overnight at 37 °C. MIC is the lowest concentration of an antibacterial agent necessary to inhibit visible growth. MBC is the minimum concentration of an antibacterial agent that results in bacterial death [26].

### 2.6. Cell Viability and Cytotoxic Effects

#### 2.6.1. Mammalian Cell Lines

Human lung cancer cell line (A549), human colon adenocarcinoma cell line (Caco-2), human breast cancer cell line (MDA-MB-231), and normal lung fibroblasts (HEL299) were used to assess the cytotoxic effects of the different extracts. These cell lines were obtained from the American Type Culture Collection (ATCC, Rockville, MD, USA) through personal communication with Prof. Dr. Mahmoud Al-Aeser, the Regional Center for Mycology and Biotechnology (RCMB) at Al-Azhar University, Egypt.

#### 2.6.2. Propagation of Cell Lines

Dulbecco’s modified Eagle’s medium (DMEM) was used to propagate the cells. The media were supplemented with 10% heat-inactivated fetal bovine serum (FBS), 1% L-glutamine, HEPES buffer, and 50 µg/mL gentamycin. All cells were maintained at 37 °C in a humidified atmosphere with 5% CO_2_ and were sub-cultured two times a week.

#### 2.6.3. Cytotoxicity Assay

The cytotoxicity assay was evaluated as described by Mosmann [27]. Briefly, the cells were seeded in a 96-well plate at a cell concentration of 1 × 10^4^ cells per well in 100 µL of growth medium using triplicate samples. A fresh medium containing different concentrations of the methanolic extract was added 24 h after seeding. Two-fold diluents of the methanolic extract were also added to confluent cell monolayers dispensed into 96-well, flat-bottomed microtiter plates (Falcon, NJ, USA) using a multichannel pipette. The microtiter plates were incubated at 37 °C in a humidified incubator with 5% CO_2_ for a period of 24 h. Control cells were incubated without extract and with or without DMSO. The DMSO concentration in each well was too low (maximal 0.1%) to affect the experiment.

After the end of the incubation period, the media in the well were aspirated and crystal violet solution (1%) was added to each well for at least 30 min. The plates were then rinsed with tap water until all the excess stains had been removed. Glacial acetic acid (30%) was subsequently added to all wells, mixed thoroughly and the plates placed into the microplate reader (SunRise, TECAN, Inc., Morrisville, NC, USA) for absorbance measurements at a test wavelength of 490 nm. All the results were corrected for the background absorbance detected in wells without added stain. Treated samples were compared with the control cell in the absence of the tested compounds. The cytotoxic effects of different concentrations of the methanolic extracts on cells were calculated as described by Mosmann [27].

The viability is expressed as percentage and the IC_50_ was calculated by using a dose–response curve for each concentration using GraphPad Prism software (San Diego, CA, USA).

### 2.7. In Vitro Anti-Inflammatory Activity

#### 2.7.1. Erythrocyte Suspension Preparation

The membrane stabilization approach was used to test the anti-inflammatory efficacy in vitro. RBCs were extracted from a healthy human volunteer who had not taken any non-steroidal anti-inflammatory medicines for two weeks before the trial. The blood was cleansed three times in a 10 mM sodium phosphate-buffered isotonic buffered solution (154 mM NaCl) (pH 7.4). The blood was centrifuged at 3000× *g* for 10 min.

#### 2.7.2. Hypotonic Solution-Induced Erythrocyte Hemolysis

Membrane stabilizing activities of the methanolic extract (with high cytotoxic activity) were assessed using hypotonic solution-induced erythrocyte hemolysis. The sample used consisted of a stock erythrocyte (RBCs) suspension (0.50 mL) mixed with 5 mL of hypotonic solution (50 mM NaCl) in 10 mM sodium phosphate-buffered saline (pH 7.4) containing the methanolic extracts (7.81–1000 µg/mL) or indomethacin (as positive control). The negative control sample consisted of 0.5 mL of RBCs mixed with hypotonic-buffered saline solution alone. The mixtures were incubated in 96 well plates for 10 min at room temperature and centrifuged for 10 min at 3000× *g* and the absorbance of the supernatant was measured at 540 nm. The percentage inhibition of hemolysis or membrane stabilization was calculated according to the modified method described by Shinde et al. [28].
Inhibition of hemolysis (membrane stabilization %) = OD_1_ − OD_2_/OD_1_ × 100
OD_1_ = Optical density of the hypotonic-buffered saline solution alone
OD_2_ = Optical density of extract in the hypotonic solution

The IC_50_ value was defined as the concentration of the sample to inhibit 50% RBCs hemolysis under the assay conditions.

### 2.8. Determination of the Phenolic Acids and Flavonoids of Solenostemma argel Methanolic Extract

HPLC (Agilent 1100, Agilent Technologies, Inc., Santa Clara, CA, USA), which was used for the detection of phenolic acids and flavonoids, consisted of two LC pumps, a UV/V detector, and a C18 column (150 × 4.60 mm, 5 µm particle size). The Agilent ChemStation was used to obtain and analyze chromatograms. Phenolic acids were separated using a gradient mobile phase consisting of two solvents: solvent A (methanol) and solvent B (acetic acid in ultrapure water, 1:25). Elution from the column was achieved with the following gradients: 0 to 3 min of solvent B, followed by 50% eluent A for the next 5 min, after which the concentration of A was increased to 80% for the next 2 min and then reduced to 50% again for the following 5 min. Flavonoids were separated using a gradient mobile phase consisting of two solvents: solvent A (acetonitrile) and solvent B (0.2%, *v*/*v* aqueous formic acid) with an isocratic elution (70:30) program. The detection wavelength was set between 200 and 450 nm, with specific monitoring conducted at 220 nm. Identification of the phenolic and flavonoid compounds was performed by comparing the retention times of the analyses with reference standards according to the methods described by Mattila et al. [29].

### 2.9. Determination of the Volatile Components of Solenostemma argel Methanolic Extract

GC-MS analysis of the methanolic extract was carried out using Thermo Scientific TRACE 1310 Gas Chromatograph (Waltham, MA, USA) coupled with an ISQ LT (single quadrupole mass spectrometer). The column was DB5-MS, 30 m, 0.25 mm ID (J&W Scientific, Folsom, CA, USA). Helium at a flow rate of 1.0 mL/min was used as carrier gas. The temperature program was as follows: started at 40 °C, sample held at 40 °C for 3 min; increasing to 280 °C with 5.0 °C/min heating rate ramp and maintained at this temperature for 5 min and before an increase to 290 °C with 7.5 °C/min heating rate and samples then maintained at this temperature for 1 min. The injection and detector temperatures were 200 and 300 °C, respectively. Mass spectra were obtained by electron ionization (EI) at 70 eV, using a spectral range of *m*/*z* 40–450. The compounds were identified by Wiley and Nist mass spectral data base.

### 2.10. Statistical Analysis

An analysis of variance (ANOVA) was performed on the data to examine any significant difference between the samples used in his study. The means of triplicate samples were calculated as well as their standard deviation (SD). Duncan’s multiple range tests (*p* ≤ 0.05) were used to examine the significance of the variable mean differences. All statistical analyses were carried out using IBM SPSS version 16 (SPSS Inc, Chicago, IL, USA).

## 3. Results and Discussion

### 3.1. DPPH Radical Scavenging Activity of Solenostemma argel Extracts

To investigate the potential health properties of *Solenostemma argel*, we carried out a screening of the antioxidant abilities of different extracts of *Solenostemma argel*. Regarding the effect of the extraction solvent on antioxidant activity, methanol extract had the highest scavenging potency at 79.36%, followed by ethanol at 66.17% and acetone at 61.77%, respectively, while chloroform showed the lowest scavenging potency at 40.34% (Table 1).

Regarding the effect of the extract’s concentration, the scavenging potency increased with the increasing *Solenostemma argel* concentration. The highest scavenging percentage 94.37% was detected using the concentration of 1280 µg/mL that was followed by 89.72% at a concentration of 640 µg/mL, and 79.31% at a concentration of 320 µg/mL, respectively. On the other hand, the concentration of 10 µg/mL gave the lowest scavenging potency with an average of 12.16%. The lowest IC_50_ value of 16.8 μg/mL was recorded with methanolic extract, while the highest value of 129.6 μg/mL was recorded with water extract (Table 2).

The observed results were consistent with the findings of Kebbab-Massime et al. [30], who found that the methanolic extract of *Solenostemma argel* showed higher radical scavenging activity than the aqueous extract at all doses evaluated using the DPPH assay. Taj et al. [31] stated that the radical scavenging activity of *Solenostemma argel* increased from 32% at a concentration of 250 µg /mL to 84% at a concentration of 1000 µg/mL. Al-Juhaimi et al. [32] stated that the antioxidant activity of argel extract was due to the presence of phenolic acids, flavones, glycosylated flavonoids, polyphenols, b-carotene, b-sitosterol, monoterpenes, pregnenes, and pregnan. Benmaarouf et al. [21] stated that the high content of flavonoids such as rutin, kaempferol-3-o-rutinoside, kaempferol-3-o-diglucoside-7-o-glucoside, astragalin, and kaempfero in *Solenostemma argel* imparted its antioxidant activity.

### 3.2. Antimicrobial Activity of Solenostemma argel Extracts

Acetone, chloroform, and ethyl acetate were the most active solvents on most isolates, whereas methylene chloride and methanol exhibited moderate activity. Water, ether, and ethanol extracts, on the other hand, exhibited no inhibitory impact on isolated bacteria. (Table 3). The most sensitive organism was *S. epidermidis* with an inhibition zone of 27 mm using chloroform extract.

Chloroform was the best extract with least MIC and MBC followed by acetone and ethyl acetate. The MIC of chloroform ranged from 6.25 to 25 mg/mL, while the MBCs of the chloroform extract ranged from 12.5 to 100 mg/mL among the tested strains. The least MIC and MBC were (6.25, 12.5) mg/mL, respectively, which were recorded with *S.*
*epidermidis* (Table 4).

Our findings support the findings of numerous studies, such as Sulieman, et al. [33], who found that *Solenostemma argel* extracts had potent antimicrobial activity against *Aspergillus niger, Pennicilium italicum*, *Escerichia coli* and *Salmonella typhi*. Tharib et al. [34] also stated that *Solenostemma argel* extract has potent antimicrobial activity against both Gram-positive and Gram-negative bacteria. According to Megeressa et al. [35] Gram-positive bacteria are more sensitive to plant extracts than Gram-negative bacteria, which is likely owing to the presence of phospholipid membranes in the cell wall of Gram-negative bacteria. Hamadnalla et al. [36] stated that the antibacterial activity is due to the presence of phytoconstituents such as saponin and flavonoid in different plant extracts.

### 3.3. Anti-Proliferative Effects on A549, Caco-2, and MDA-MB-231 Cells

IC_50_ of *Solenostemma argel* methanolic extract against the lung carcinoma cell line (A549), Caco-2 cell line and MDA-MB-231 cell line were (179, 243 and 731), µg/mL, respectively. While IC_50_ of Doxorubicin (reference standard) against the lung carcinoma cell line (A549), Caco-2 cell line, and MDA-MB-231 cell line were (0.95, 1.93 and 0.6), µg/mL, respectively (Table 5).

Abouzaid et al. [37] found that the methanolic extract of *Solenostemma argel* had significant cytotoxic effects on lung cancer induced by dimethylbenzanthracene (DMBA) in male Wistar rats. Hanafi and Mansour [38] investigated the anticancer activity of aqueous extract of *Solenostemma argel* against Ehrlich carcinoma-bearing mice. It was found that the *Solenostemma argel* aqueous extract activated tumor cell death, and decreased tumor volume. It also recorded a high and wide zone of apoptotic tumor cells.

### 3.4. In Vitro Anti-Inflammatory Activity (Membrane Stabilization %) of Solenostemma argel Methanolic Extract

At various concentrations (1000, 500, 250, 125, 62.5, 31.25, 15.63, and 7.8 g/mL), the methanolic extract of *Solenostemma argel* demonstrated significant stabilization of RBC membranes. It was observed that the percentage protection of methanolic extract of *Solenostemma argel* increased from 28.12% at a concentration of 7.8 µg/mL to 82.73% at a concentration of 1000 µg/mL. The IC_50_ value of methanolic extract of *Solenostemma argel* was 24.4 ± 0.96 µg/mL compared to 17.02 ± 0.91 µg/mL with indomethacin standard. (Table 6).

Innocenti et al. [39] evaluated the anti-inflammatory activity of *Solenostemma argel* using the Croton oil ear test in mice. The extract induced 73% oedema reduction compared to 56% reduction induced by indomethacin. Ibrahim et al. [40] anti-inflammatory activity (74.19%, 69.44% and 66.58%), respectively, by inhibiting the heat-induced albumin denaturation. Ismaiel et al. [41] reported that *Solenostemma argel* extract exhibited inhibition of inflammation, at a dose of 100 mg/kg body weight after 3 h (22%) of carrageenan administration. The presence of flavonoids and related polyphenols in *Solenostemma argel* extract may be responsible for its anti-inflammatory activity [39]. Benmaarouf et al. [21] stated that the anti-inflammatory activity of *Solenostemma argel* may be due to the inhibition of the release of anti-inflammatory mediators occurring during the intermediate and second phases of edema formation, such as bradykinin and prostaglandins.

### 3.5. Identification of Phenolic Acids and Flavonoids of Solenostemma argel Methanolic Extract

Table 7 and Figure 1A,B presented the different phenolic compounds obtained from the analyses of *Solenostemma argel* methanolic extract by HPLC. Data indicated the presence of a total of eleven different phenolic compounds consisting of five flavonoids (rutin, quercetin, kaempferol, luteolin, and catechin) and five phenolic acids (syringic acid, p-coumaric acid, caffeic acid, gallic acid, and ferulic acid).

Data indicated that the most abundant phenolic compound was gallic acid at 36.49%, then synergic acid and p-coumaric acid were 18.28% and 18%, respectively. The most abundant flavonoid compound was catechin at 47.32%, then quercetin at 18.47%, then luteolin at 17.59%, then kaempferol at 14.36%, and the lowest amount of flavonoids was rutin at 2.24%.

The results obtained were consistent with those reported by Al-Juhaimi et al. [32] who found that gallic acid, caffeic acid, ferulic acid, and syringic acid are the primary phenolic acids and kaempferol, quercetin, and catechin are the common flavonoids present in methanolic extract of *Solenostemma argel*. Benmaarouf et al. [21] demonstrated the presence of salicylic acid, rutin, kaempferol-3-*O*-rutinoside, kampeferol 3-*O*-diglucoside-7-*O* glucoside in acetone extract of *Solenostemma argel*. Ibrahim et al. [40] reported the presence of other phenolic compounds such as pyrogallol, benzoic acid, resveratrol, and other flavonoids such as hesperetin, apigenin, and hesperidin in the methanolic extract of *Solenostemma argel*.

### 3.6. Determination of the Volatile Components of Solenostemma argel Methanolic Extract

Table 8 and Figure 2 demonstrated the volatile components of a methanolic extract of *Solenostemma argel* using GC-MS. Data in Table 8 and Figure 2 indicated that nineteen bioactive substances were identified and categorized based on their chemical structures using GC-MS analysis. cis-2,6-dimethyl-2,6-octadiene (8.95%), 2,6-octadiene, 2,6 dimethyl (4.19%), cyclohexene, 1-methyl-4-(1 methylethenyl)-,(S) (2.70%), 1,3-dioxolane,4-methyl-2-pentadecyl (9.69%), bicyclo [3.1.0] hexane, 4-methylene-1-(1-methylethyl) (1.64%), *N*,*N*′-bis (3-Aminopropyl) ethylenediamine (3.33%), 2-furanmethanol, 5-ethenyltetrahydro-À,À,5-trimethyl-, cis (5.01%) 2-furanmethanol,5-ethenyltetrahydro-À,À,5-trimethyl-, cis (4.41%), linalool (2.87%) nonanoic acid, 9-oxo-, methyl ester (3.33%), methyl 3-methylbutanoate (7.34%), hexanoic acid, methyl ester (10.93%), cis-3-hexenyllactate (2.97%), phenol, 2-(1,1-dimethylethyl) (8.50%), 2-(5-methyl-5 vinyltetrah ydro-2-furanyl)-2-propanol (11.63%), 1,3,5-triazine-2,4-diamine,6-chloro-n-ethyl (3.12%), hexadecanoic acid, methyl ester (4.80%), 9,12-octadecadienoic acid (z,z)-, methyl ester (1.85%), 10-octadecenoic acid, methyl ester (2.76%), and 2-(5-methyl-5 vinyl tetrahydro-2-furanyl)-2-propanol (11.63%) were the main volatile components.

El-Sonbat et al. [42] recorded twenty-two volatile organic compounds within the methanolic extract of argel, which are xylitol (35.65%), 4- methylcatechol (23.89%), sinapic acid (8.13%), linalool acetate (4.28%), α-Copaene (3.21%), 2- (but-2’-enyl) phenol (2.85%), menthol,1’-(butyn3-one-1-yl)-, (1S,2S,5R)- (2.85%), α-Elemene (2.14%), and junipene (2.14%), were found to be the major constituents, while 7-methoxy-3-(4- methoxyphenyl)-2,3-dihydro-4H-chromen-4-one (2.14%), β-Guaiene (1.6%); β-Asarone (1.6%), phenol, 4-(3-hydroxy-1-propeenyl)-2-methoxy- (1.5%), isoferulic acid (1.43%), γ-Himachalene (1.25%), L-aspartic acid (1.07%), resorcinol (0.71%), valproic acid (0.71%), 6- monohydroxyflavone (0.53%), tricosanoic acid (0.53%), apigenin-7-glucoside (0.53%), 3,7,3’,4’- tetra-*O*-methyl-5-*O*-(trimethylsilyl) quercetin (0.36%), phenol, 2-methoxy-5-(1-propenyl)- (0.36%), α-Terpineol (0.36%), and 6- hydroxyflavone (0.18%) were the minor constituents.

## 4. Conclusions

The present study concludes that *Solenostemma argel* contains considerable quantities of phenolic compounds, flavonoids, and volatile compounds. *S. argel* is a very promising source of novel non-toxic, anti-inflammatory, and antioxidant compounds. It can be also used in the food industry as an antimicrobial agent. These results also provide a scientific basis for its use in cancer treatment. However, further studies are necessary to find active components in *Solenostemma argel* extract and to confirm its mechanism of action.

## Figures and Tables

**Figure 1 molecules-27-08118-f001:**
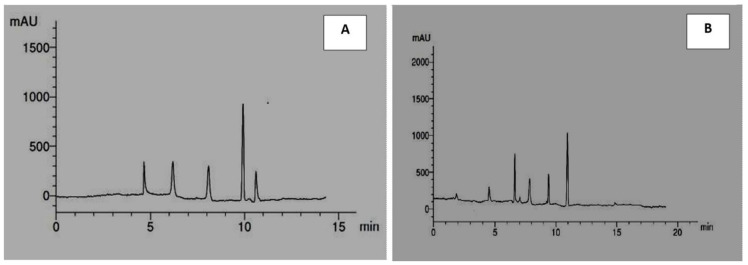
HPLC chromatographic profiles of phenolic acids (**A**) and flavonoids (**B**) in methanolic extract of *Solenostemma argel*.

**Figure 2 molecules-27-08118-f002:**
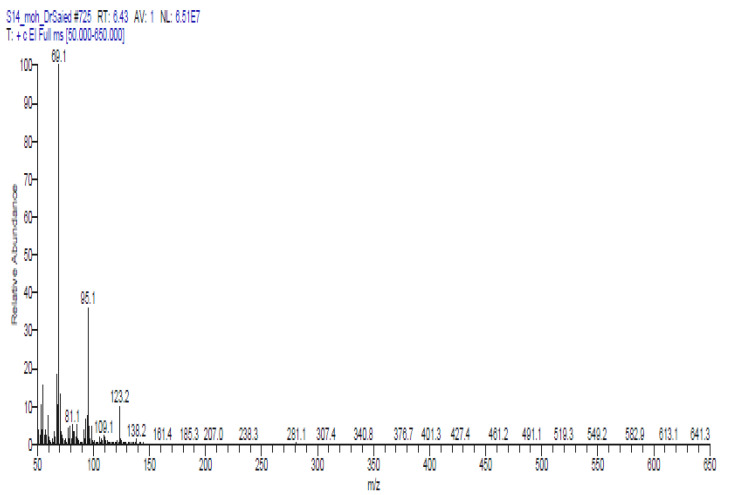
GC-MS chromatographic for identification of compounds in methanolic extract of *Solenostemma argel*.

**Table 1 molecules-27-08118-t001:** Radical scavenging activity of *Solenostemma argel extracts* at different concentrations toward DPPH.

Conc. (µg/mL)	* Radical Scavenging Activities of Extracts								
	Water	Acetone	Chloroform	Methylene Chloride	Ether	Methanol	Ethanol	Ethyl Acetate	*Group Mean ± S.E*
1280	96.53 ± 0.79	94.78 ± 0.86	93.84 ± 0.92	96.82 ± 0.64	94.59 ± 1.07	98.47 ± 0.59	95.06 ± 0.62	89.94 ± 0.78	94.37 ± 0.78 ^a^
640	91.65 ± 1.23	91.36 ± 0.72	82.59 ± 1.37	90.96 ± 1.32	86.24 ± 1.83	96.29 ± 0.63	92.82 ± 0.74	85.86 ± 1.32	89.72 ± 1.14 ^b^
320	76.24 ± 1.48	85.16 ± 1.34	57.06 ± 2.86	82.24 ± 1.78	78.47 ± 1.59	94.65 ± 0.41	89.41 ± 0.65	71.29 ± 1.53	79.31 ± 1.45 ^c^
160	56.98 ± 1.76	76.35 ± 1.97	37.88 ± 3.94	67.06 ± 1.92	60.59 ± 2.65	92.12 ± 0.86	78.57 ± 1.33	55.53 ± 2.71	65.63 ± 2.13 ^d^
80	38.62 ± 3.44	55.43 ± 2.81	21.76 ± 1.83	50.76 ± 1.89	42.71 ± 3.13	87.35 ± 0.97	67.75 ± 1.44	43.18 ± 2.84	50.94 ± 2.29 ^e^
40	25.76 ± 1.28	40.88 ± 2.76	16.24 ± 1.62	31.88 ± 2.36	29.70 ± 2.34	74.06 ± 1.42	53.62 ± 1.94	27.65 ± 1.97	37.47 ± 1.96 ^f^
20	18.56 ± 0.73	28.41 ± 2.93	10.43 ± 0.81	20.82 ± 1.42	13.94 ± 1.52	57.18 ± 2.98	31.48 ± 2.36	19.42 ± 1.06	25.03 ± 1.72 ^g^
10	7.65 ± 0.19	15.82 ± 1.74	2.94 ± 0.54	14.58 ± 0.78	4.35 ± 0.61	34.47 ± 3.15	19.76 ± 1.42	7.76 ± 0.82	12.16 ± 1.15 ^h^
*Group mean ± S.E*	51.49 ± 1.24 ^e^	61.77 ± 1.77 ^c^	40.34 ± 1.54 ^g^	56.81 ± 1.54 ^d^	51.32 ± 2.23 ^f^	79.36 ± 1.12 ^a^	66.17 ± 1.52 ^b^	50.1 ± 1.75 ^f^	

* Radical scavenging activity given as percentage inhibition. The percentage inhibition value of the standard compound ascorbic acid was 99.23% at a concentration of 1280 µg/mL. Means followed by different letter (s) within a column or row are significantly different (*p* ≤ 0.05) according to Duncan’s multiple range test.

**Table 2 molecules-27-08118-t002:** IC_50_ values of *Solenostemma argel* extracts.

Type of Extract	IC_50_ (µg/mL)
Water	129.60 ± 5.48
Acetone	65.10 ± 3.1
Chloroform	26.10 ± 17.4
Methylene chloride	78.40 ± 2.84
Ether	112.60 ± 4.6
Methanol	16.80 ± 0.62
Ethanol	36.70 ± 3.2
Ethyl acetate	124.20 ± 6.4

The IC_50_ values of *Solenostemma argel* extracts were calculated using Quest Graph™ IC_50_ Calculator (AAT Bioquest, Inc., Sunnyvale, CA, USA). IC_50_ was determined with a non-linear model.

**Table 3 molecules-27-08118-t003:** Antimicrobial activity of different *Solenostemma argel* extracts against respiratory tract infection isolates measured as zone of inhibition diameter (ZOI, mm).

Isolates	Extracts							
	Water	Methanol	Acetone	Ethanol	Chloroform	Ether	Ethyl Acetate	Methylene Chloride
*S. aureus*	NI **	16.5 ± 2.1 *	22.5 ± 3.5	NI	22.5 ± 0.7	NI	24.5± 0.7	15 ± 0
*S. epidermidis*	NI	16 ± 1.4	18.5 ± 2.1	NI	27 ± 0	NI	19 ± 0	NI
*E. faecalis*	NI	14 ± 1.4	13.5 ± 0.7	NI	NI	NI	NI	NI
*E. coli*	NI	19.5 ± 3.5	23 ± 0	NI	21.5 ± 2.1	NI	19.5 ± 3.5	14 ± 1.4
*E. cloacae*	NI	18.5 ± 2.1	23.5 ± 2.1	NI	17.5 ± 0.7	NI	23.5 ± 0.7	14.5 ± 0.7
*P. aeruginosa*	NI	17.5 ± 0.7	23 ± 2.8	NI	22.5 ± 2.1	NI	24.5 ± 0.7	13.5 ± 0.7
*A. baumannii*	NI	16 ± 1.4	23 ± 0	NI	21 ± 1.4	NI	22.5 ± 3.5	15 ± 0
*C. tropicalis*	NI	17.5 ± 3.5	23 ± 1.4	NI	19.5 ± 0.7	NI	26 ± 1.4	15 ± 0

* Zone of inhibition (ZOI) diameters included the diameter of paper disc (6 mm) ± standard deviations. ** NI: no inhibitory effect.

**Table 4 molecules-27-08118-t004:** Minimum inhibitory concentration (MIC) and minimum bactericidal concentration (MBC) for different extracts of *Solenostemma argel* against respiratory tract infections microbes.

Isolates	Extracts					
	(MIC) * (mg/mL)			(MBC) ** (mg/mL)		
	Acetone mg/mL	Chloroform mg/mL	Ethyl Acetate mg/mL	Acetone mg/mL	Chloroform mg/mL	Ethyl Acetate mg/mL
*S. aureus*	12.5	25	25	50	100	100
*S. epidermidis*	12.5	6.25	25	50	12.5	25
*E. faecalis*	25	N	N	100	N	N
*E. coli*	25	25	25	100	25	100
*E. cloacae*	25	25	25	100	25	100
*P. aeruginosa*	25	25	12.5	100	25	50
*A. baumannii*	25	25	25	25	25	25
*C. tropicalis*	25	25	25	25	100	100

* MIC: minimum inhibitory concentration, ** MBC: minimum bactericidal concentration.

**Table 5 molecules-27-08118-t005:** Anticancer activity of *Solenostemma argel* methanolic extract at different concentrations toward different cancer cell lines.

Samples Conc. (µg/mL)	*Solenostemma argel* Methanol Extract								
	Cancer Cell Lines								
	^#^ A-549			^##^ CACO_2_			^###^ MDA-MB-231		
	Viability (%)	Inhibitory (%)	S.D. (±)	Viability (%)	Inhibitory (%)	S.D. (±)	Viability (%)	Inhibitory (%)	S.D. (±)
1000	7.83	92.17	1.09	13.95	86.05	1.09	34.59	65.41	2.37
500	21.67	78.33	1.75	32.76	67.24	3.42	63.73	36.73	3.91
250	37.85	62.15	3.49	48.52	51.48	2.86	80.95	19.05	2.83
125	59.21	40.79	2.37	76.31	23.96	3.97	93.83	6.17	1.95
62.5	78.04	21.96	1.78	88.42	11.58	2.64	99.56	0.44	0.48
31.25	95.26	4.74	0.82	97.16	2.84	1.32	100	0	0
15.6	99.13	0.78	0.51	100	0	0	100	0	0
7.8	100	0	0	100	0	0	100	0	0
0	100	0	0	100	0	0	100	0	0

^#^ A549 cell line: lung carcinoma cell, ^##^ CACO_2_: intestinal carcinoma cell line, ^###^ MDA-MB-231; breast cancer cell line.

**Table 6 molecules-27-08118-t006:** Anti-inflammatory activity (membrane stabilization %) of *Solenostemma argel methanolic extract* at different concentrations.

Samples Concentration (µg/mL)	Anti-Inflammatory Activity (Membrane Stabilization %)			
	*Solenostemma argel Methanolic Extract*		Indomethacin (as Positive Control)	
	Membrane Stabilization (%)	S.D.	Membrane Stabilization (%)	S.D.
1000	82.73	2.1	100	0
500	76.44	0.63	95.35	0.63
250	71.35	0.68	78.34	2.1
125	68.47	0.58	72.35	0.58
62.5	62.95	2.1	68.35	1.5
31.25	57.95	1.6	56.38	1.3
15.6	39.96	1.4	49.38	0.72
7.8	28.12	0.94	41.18	1.3
0	0	0	0	0
IC_50_	24.4 ± 0.96		17.02 ± 0.91	

All determinations were carried out triplicate manner and values are expressed as *mean ± S.E*. The IC_50_ value is defined as the concentration of inhibitor 50% of its activity under the assayed conditions.

**Table 7 molecules-27-08118-t007:** Phenolic acids and flavonoid compositions of methanolic extract of *Solenostemma argel*.

No.	Retention Time (min)	Compound Name	Peak Area%
1	4.8	Syringic acid	5.22
2	6.0	P-coumaric acid	5.14
3	8.0	Caffeic acid	4.69
4	10.0	Gallic acid	10.42
5	10.9	Ferulic acid	3.08
6	4.8	Rutin	0.59
7	6.8	Quercetin	4.86
8	8.0	Kaempferol	3.87
9	9.1	Luteolin	4.63
10	11.0	Catechin	12.45

**Table 8 molecules-27-08118-t008:** Volatiles composition of methanolic extract of *Solenostemma argel*.

No.	Compound Name	Retention Time (min)	Molecular Formula	*m*/*z* Fragments	Peak Area Percentage #
1	Cis-2,6-dimethyl-2,6-octadiene	6.43	C_10_H_18_	41, 53, 69 *, 81, 95, 109, 123, 138	8.95
2	2,6-octadiene, 2,6 dimethyl-	6.73	C_10_H_18_	27, 41, 53, 69 *, 81, 95, 109,128, 138	4.19
3	cyclohexene, 1-methyl-4-(1 methylethenyl)-, (S)-	7.07	C_10_H_16_	27, 41, 53, 68 *, 79, 93, 107, 121, 136	2.70
4	1,3-dioxolane,4-methyl-2-pentadecyl	7.34	C_19_H_38_O_2_	30,43, 46, 57 *, 69, 81, 87, 105	9.69
5	Bicyclo[3.1.0]hexane, 4-methylene-1-(1-methylethyl)	7.84	C_10_H_16_	41, 69, 77, 93 *, 105, 121, 121	1.64
6	*N*,*N*′-bis(3-aminopropyl) ethylenediamine	8.12	C_8_H_22_N_4_	30, 44 *, 58, 71, 87, 100, 154, 175	3.33
7	2-furanmethanol, 5-ethenyltetrahydro-à,à,5-trimethyl-, cis	8.44	C_10_H_18_O_2_	43, 55, 59 *, 68, 81,94, 111, 155	5.01
8	2-furanmethanol,5-ethenyltetrahydro-à,à,5-trimethyl-, cis	8.85	C_10_H_18_O_2_	43, 59 *, 68, 81, 94, 111, 155	4.41
9	linalool	9.23	C_10_H_18_O	41, 55, 71 *, 80, 93, 107, 121, 136	2.87
10	Nonanoic acid, 9-oxo-, methyl ester	10.99	C_10_H_18_O_3_	55, 69, 74 *, 87, 100, 127, 143, 186	3.33
11	Methyl 3-methylbutanoate	11.19	C_8_H_16_O_5_	41, 57, 69, 74 *, 85, 101, 116	7.34
12	Hexanoic acid, methyl ester	11.77	C_7_H_14_O_2_	18, 29, 43, 55, 59, 74 *, 87, 99,130	10.93
13	Cis-3-hexenyllactate	12.34	C_9_H_16_O_3_	45, 55, 67, 82 *, 89, 99, 141, 157, 172	2.97
14	Phenol, 2-(1,1-dimethylethyl)-	15.31	C_10_H_14_O	65, 77, 91, 107, 115, 135, 135 *, 150	8.50
15	2-(5-methyl-5 vinyl tetrahydro-2-furanyl)-2-propanol	17.82	C_10_H_18_O_2_	43, 59 *, 68, 94, 111, 137, 155	11.63
16	1,3,5-triazine-2,4-diamine, 6-chloro-n-ethyl	27.44	C_5_H_8_ClN_5_	43 *, 55, 71, 85, 97, 111, 125, 125, 145, 158, 173	3.12
17	Hexadecanoic acid, methyl ester	29.10	C_17_H_34_O_2_	43, 74 *, 87, 97, 129, 143, 185, 227, 270	4.80
18	9,12-octadecadienoic acid (z,z)-, methyl ester	32.33	C_19_H_34_O_2_	67 *, 81, 95, 220, 293, 294	1.85
19	10-octadecenoic acid, methyl ester	32.44	C_19_H_36_O_2_	41, 55 *, 69, 111, 180, 222, 264, 296	2.76

* The most intense ion is indicated by the base peak (tallest peak). # (Peak area of individual volatile compound/total peak areas of all volatiles) × 100.

## Data Availability

Not applicable.

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
