# Peer review of "Pharmacological Activities and Characterization of Phenolic and Flavonoid Compounds in Solenostemma argel Extract"

_molecules, 2022, doi:10.3390/molecules27238118_

Round 1

Reviewer 1 Report

The work is interesting and the results are scientifically significant. The authors can find the comments in the PDF attached. 

Author Response

Dear reviewer

I replied to all comments in your pdf and attached i

Many thanks for your valuable comments

Reviewer 2 Report

This study is interesting because it contributes to evaluate the pharmacological activities and the characterization of phenolic and flavonoid compounds in extracts of Solenostemma argel.

The Methodology of the assays and the Results are very complete and well described, presented with two figures and eight tables, and the most important findings are well connected and discussed with references to other studies in the Discussion.

The results revealed that S. argel methanolic extracts contain considerable quantities of phenolic compounds, flavonoids, and volatile compounds. It´s a promising source of novel nontoxic anti-inflammatory, and antioxidant compounds. It can also be used as antimicrobial and anticancer agents.

There are some minor changes that need to be made. The proposed review changes are as follows:

- On page 2, line 22, the name of the species should be with the descriptor, like thisSolenostemma argel (Delile) Hayne”;

- On page 3, lines 129, 130 and 131, the names of the species should be in italic.

With these changes, the recommendation will be to accept the manuscript for publication.

Author Response

Reviewer 2

This study is interesting because it contributes to evaluate the pharmacological activities and the characterization of phenolic and flavonoid compounds in extracts of Solenostemma argel.

The Methodology of the assays and the Results are very complete and well described, presented with two figures and eight tables, and the most important findings are well connected and discussed with references to other studies in the Discussion.

The results revealed that S. argel methanolic extracts contain considerable quantities of phenolic compounds, flavonoids, and volatile compounds. It´s a promising source of novel nontoxic anti-inflammatory, and antioxidant compounds. It can also be used as antimicrobial and anticancer agents.

There are some minor changes that need to be made. The proposed review changes are as follows:

- On page 2, line 22, the name of the species should be with the descriptor, like this “Solenostemma argel (Delile) Hayne”;

Thanks for reviewer’s comments

The name of species has been modified in the revised manuscript

- On page 3, lines 129, 130 and 131, the names of the species should be in italic.

Thanks for reviewer’s comments.

The names of the species made in italic in the revised manuscript.

With these changes, the recommendation will be to accept the manuscript for publication.
